# Evaluation of Sebostatic Activity of *Juniperus communis* Fruit Oil and *Pelargonium graveolens* Oil Compared to Niacinamide

**Justyna Kozlowska \*, Anna Kaczmarkiewicz, Natalia Stachowiak and Alina Sionkowska**

Faculty of Chemistry, Nicolaus Copernicus University in Torun, Gagarin 7, 87-100 Torun, Poland; 263215@stud.umk.pl (A.K.); 258892@stud.umk.pl (N.S.); as@chem.umk.pl (A.S.)

\* Correspondence: justynak@chem.umk.pl; Tel.: +48-56-6114833

**Abstract:** As a facial skin condition, oily skin causes cosmetic problems, such as large pores, shiny appearance, and the feeling of greasiness and heaviness. Furthermore, extensive sebum production leads to common skin disorders such as acne vulgaris or seborrheic dermatitis. This study investigated the efficacy of sebum control tonics containing *Juniperus communis* fruit oil, *Pelargonium graveolens* oil, or niacinamide. The effects of *Juniperus communis* fruit oil, *Pelargonium graveolens* oil, and niacinamide on sebum excretion rates were investigated using Sebumeter®. Sebum measurements (Sebumeter® SM 815, Courage & Khazaka®, Köln, Germany) were made on the skin surface in three places by applying the sebumeter probe to the forehead after 10, 60, and 120 min from application of the tonic. The results indicated that the application of the tonic maintained a lower sebum secretion 10 min and 60 min after the application of the cosmetic, compared to those before it. However, a visible sebum-reducing efficacy after 2 h was reported only for tonic containing 0.25% *Pelargonium graveolens* oil and for the tonic with the addition of 3% niacinamide. After 2 h, the values of sebum measurements were $44 \pm 5.13$ a.u. and $58 \pm 9.07$ a.u., respectively. Our results show that the tonic with the addition of 0.25% *Pelargonium graveolens* oil is the most effective in reducing sebum production.

**Keywords:** sebum; sebostatic activity; essential oil; niacinamide

## 1. Introduction

Sebaceous glands are part of a hair-sebum apparatus and grow with the increase of a hair follicle. They are placed all over skin (except for the inner part of the hand and the bottom of the feet) and their proliferation and size are characteristic of body parts. Most of the sebaceous glands are in seborrheic areas: on the face in the T-zone, on the chest, and on the back [1]. On the face, there are between 300–1500 glands/$cm^2$; on the scalp there are about 300–500 glands/$cm^2$; and in other seborrheic areas there are 100 glands/$cm^2$ or less. They are identified by three different types of sebaceous follicles, according to the size of the sebaceous glands [2].

Human sebaceous glands are holocrine-secreting tissues. The most primary function of sebaceous gland is to secrete sebum. Sebum, which is secreted to the surface of skin, contains mono-, di-, and triglycerides; free fatty acids; waxes; squalene; and cholesterol esters [3]. The hydrolipid coat on the skin surface, which is formed by mixing sebum with the secretion of sweat glands and lipids of *stratum corneum*, is the outer protecting layer for skin and hair. It prevents transepidermal water loss and protects against mechanical damage. Sebum is also responsible for controlling moisture and protecting skin from microbiological infections [4].

The activity of sebaceous glands is age-related. The sebaceous glands are most active during the foetal life and in the first months after birth, which is caused by hormones found in breastmilk. In the

next periods of life, sebaceous glands' activity drops dramatically and then increases significantly during adolescence. During perimenopause, a gradual decrease of sebum secretion among women can be observed. The most significant decrease appears at the age of 45–50 years. For men, the process of decreased sebum secretion is much slower—a small decrease can be observed only at the age of 50–55 years [5].

Overproduction of sebum is very common and results in an undesirable oily, shiny complexion with enlarged pores [3,6,7]. Oily skin is shiny in the T-zone (forehead, chin, and nose) and has a yellow-grey colour, accompanied by hyperpigmentation. It is also thick and less susceptible to irritation. Oily skin is also a source of common skin disorders such as acne vulgaris and seborrheic dermatitis [5,6,8]. For these reasons, control over excessive oiliness is very important.

There have been many attempts to develop a methodology to measure the amount of sebum secretion on the surface of the skin. The first such attempt was made in 1961. The gravimetric method used cigarette paper to collect sebum from the face. Then sebum was extracted from the tissue using a solution of an ether and then, after evaporation of the solvent, the sebum was weighed. For collecting sebum from the skin surface, bentonite clay was also used. However, the gravimetric test is very difficult and time-consuming, and therefore it is not currently used [5].

Nowadays, measuring the amount of sebum on skin and hair surface is based on the photometric method. The instrument used to measure the level of sebum is called a sebumeter (Courage & Khazaka, Germany). The sebumeter is widely used in both cosmetic and medical research [9]. The cassette (probe) of sebumeter contains a synthetic mat strip. After pressing to the surface of the skin or hair, the tape becomes transparent as a result of absorbing sebum from the surface of the measured area. Then, the tape is inserted into the socket of the measuring apparatus, which conducts a photometric examination of the extent of light transmission through the head, which corresponds to the amount of absorbed sebum [9,10].

Cosmetics for oily skin should have multi-directional effects, such as: sebostatic (inhibition of excessive amounts of sebum), bacteriostatic (limiting the growth of bacteria *Propionibacterium acnes*), inhibition of the oxidation of lipids, soothing irritations (skin irritation increases seborrhoea), keratolytic action, and a mattifying effect (absorbing sebum from the surface) [11].

Vitamin B3 (from the group of B vitamins soluble in water) can be found under different names such as vitamin PP, niacinamide, or nicotinic acid amide. Deficiency of this vitamin leads to skin lesions, called pellagra (from the Italian word for "rough skin"). Lean meat and yeast are an extensive vitamin B3 source [12]. Vitamin B3 is used in the treatment of various skin diseases because it regulates moisture, exfoliates the skin, and eliminates inflammation. Niacinamide reduces the excessive production of sebum and reduces the size of the pores of the skin [11–13].

*Pelargonium graveolens* oil is obtained from the above-ground parts of geranium plants. Pelargoniums are relatively winterhard shrubs with a height from 40 to 100 cm. All parts of the plant are covered with glandular hairs containing essential oil. It smells much like rose oil and is much cheaper, so it is commonly used for the manufacture of perfumes and the aromatization of cosmetics. Geranium essential oil is also used in medicine, because of its very strong antiseptic properties. It is also used for oily skin care [14]. It purifies and smooths the skin or adds blushes to pale complexions [15].

*Juniperus communis* fruit oil is obtained from juniper berries. They contain from 1% to 3% of essential oil. The oil is obtained from minced juniper berry by steam distillation, but the efficiency of this process is very low. Juniper essential oil extracted by this method is a yellowish or greenish liquid with a bitter taste and an intense herbal aroma. The chemical composition of juniper oil contains 120 components. Juniper essential oil is widely used, for example, in the production of alcoholic beverages, as a spice, as well as in medicine [16]. Dermatological preparations for purulent variety of acne and other skin diseases contain juniper oil. It has antibacterial and antifungal properties. It helps in the treatment of acne, dandruff, and seborrhoea. In the cosmetics industry, it is used in the manufacture of lotions and shaving preparations [15].

The aim of this work was to compare the sebostatic activity of cosmetics containing *Juniperus communis* fruit oil, *Pelargonium graveolens* oil, and niacinamide.

## 2. Materials and Methods

### 2.1. Formulation and Preparation of Tonics

Five tonics containing *Juniperus communis* fruit oil (Avicenna, Poland), *Pelargonium graveolens* oil (Avicenna, Poland), or niacinamide (Sigma Aldrich, Poznan, Poland) were prepared (Tables 1–5).

**Table 1.** Ingredients of tonic without sebostatic ingredients.

| Component (INCI [1]) | Percentage (%) |
| --- | --- |
| Aqua | 95.0 |
| Glycerin | 2.25 |
| Aloe Vera | 1.00 |
| Phenoxyethanol, Ethylhexylglycerin | 1.00 |
| Polysorbate 80 | 0.75 |

[1.] INCI: International Nomenclature of Cosmetic Ingredients.

**Table 2.** Ingredients of tonic containing 0.25% of *Juniperus communis* fruit oil.

| Component (INCI) | Percentage (%) |
| --- | --- |
| Aqua | 94.75 |
| Glycerin | 2.25 |
| Aloe Vera | 1.00 |
| Phenoxyethanol, Ethylhexylglycerin | 1.00 |
| Polysorbate 80 | 0.75 |
| *Juniperus communis* fruit oil | 0.25 |

**Table 3.** Ingredients of tonic containing 0.25% of *Pelargonium graveolens* oil.

| Component (INCI) | Percentage (%) |
| --- | --- |
| Aqua | 94.75 |
| Glycerin | 2.25 |
| Aloe Vera | 1.00 |
| Phenoxyethanol, Ethylhexylglycerin | 1.00 |
| Polysorbate 80 | 0.75 |
| *Pelargonium graveolens* oil | 0.25 |

**Table 4.** Ingredients of tonic containing 1% of niacinamide.

| Component (INCI) | Percentage (%) |
| --- | --- |
| Aqua | 94.0 |
| Glycerin | 2.25 |
| Aloe Vera | 1.00 |
| Phenoxyethanol, Ethylhexylglycerin | 1.00 |
| Polysorbate 80 | 0.75 |
| Niacinamide | 1.00 |

**Table 5.** Ingredients of tonic containing 3% of niacinamide.

| Component (INCI) | Percentage (%) |
| --- | --- |
| Aqua | 92.0 |
| Glycerin | 2.25 |
| Aloe Vera | 1.00 |
| Phenoxyethanol, Ethylhexylglycerin | 1.00 |
| Polysorbate 80 | 0.75 |
| Niacinamide | 3.00 |

*2.2. Sebum Measurement Technique*

All measurements were carried out on six adults (three women, aged 23, 24, and 25 years; and three men, aged 16, 22, and 26 years). None of the participants had taken antibiotics or retinoids or hormonal therapy at least six months before testing. The skin types of participants were normal or oily. The measurements of sebum production on the forehead, cheek, and forearm were performed using a Sebumeter® SM 815 (Courage & Khazaka®, Köln, Germany).

Make-up removal was performed before measurements using commercial micellar water without active ingredients regulating the function of sebaceous glands. After 10 min, one of the prepared tonics was applied on the forehead. Sebum measurements were taken on the skin surface in three places on each person by applying the sebumeter probe to the forehead after 10, 60, and 120 min from application of the tonic. The results of measurements were averaged and standard deviation was calculated. All measurements were performed in the laboratory in controlled temperature and humidity conditions (20–22 °C, relative humidity 40–60%).

## 3. Results and Discussion

*3.1. The Results of Sebum Measurements on the Forehead, Cheek, and Forearm*

The results of sebum measurements on the forehead, cheek, and forearm are shown in Table 6. Based on the results shown in Table 6, the forehead was chosen as the measurement area. The level of sebum on the forehead is much higher than that on cheek or forearm. This is due to the fact that the largest sebaceous glands and the greatest density of glands are found on the face, especially on the forehead [17].

**Table 6.** The results of sebum measurements on the forehead, cheek, and forearm.

| Part of Body | Measurement Values * (a.u.) |
| --- | --- |
| Forehead | $138 \pm 6.00$ |
| Cheek | $48.53 \pm 4.29$ |
| Forearm | 0 |

* **Units:** Sebumeter® units from 0 to 350 (approximated to $\mu g/cm^2$ in a certain range).

*3.2. The Result of Sebum Measurements on the Forehead after Make-Up Removal*

The result of sebum measurements on the forehead before and after make-up removal are shown in Figure 1. After analysing the results, it can be stated that make-up removal on the forehead significantly reduces the level of sebum on the skin. The sebum is removed from the skin surface, but its secretion is not inhibited.

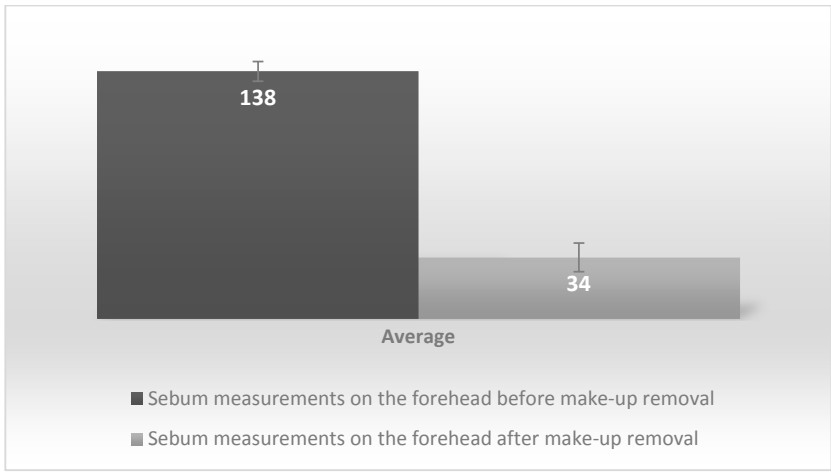

**Figure 1.** Comparison of sebum measurement results on the forehead, before and after make-up removal.

### 3.3. The Results of Sebum Measurements on the Forehead after Applying a Tonic without Sebostatic Ingredients

Table 7 shows the results of sebum measurements of the forehead which were carried out after 10, 60, and 120 min after the application of the tonic without sebostatic substances in the formulation.

**Table 7.** The results of sebum measurements on the forehead after applying the tonic without sebostatic ingredients.

| Time (min) | Measurement Values * |
|---|---|
| 10 | 45 ± 3.86 |
| 60 | 72 ± 3.94 |
| 120 | 109 ± 8.49 |

* **Units:** Sebumeter® units from 0–350 (approximated to $\mu g/cm^2$ in a certain range).

The level of sebum on the forehead after applying the tonic without sebostatic ingredients increased slightly compared to the level of sebum on the forehead after the make-up removal.

### 3.4. The Results of Sebostatic Measurements on the Forehead after Applying the Tonic with 0.25% of Essential Oils

The comparison of average values of sebum measurements after application of tonics with essential oils is shown in Figure 2.

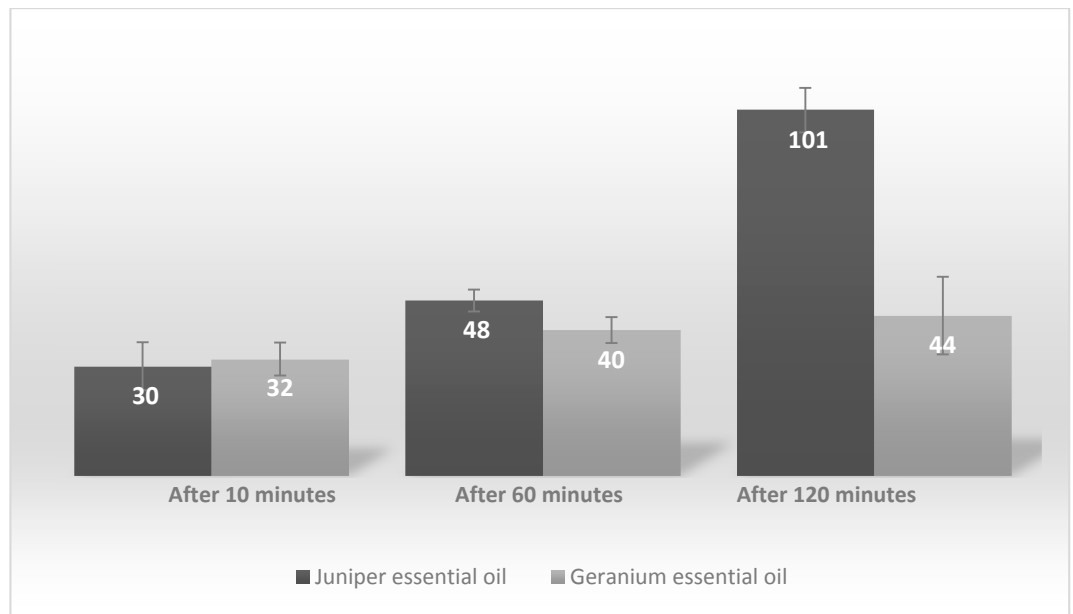

**Figure 2.** Comparison of sebum measurements on the forehead after the application of the tonic with 0.25% of *Juniperus communis* fruit oil, and the tonic with 0.25% of *Pelargonium graveolens* oil.

The results suggest that the *Juniperus communis* fruit oil has initially slightly better sebostatic activity than *Pelargonium graveolens* oil (the values of sebum measurements are 30 ± 2.72 a.u. and 32 ± 4.55 a.u., respectively). However, after 60 min, an increase of the level of the sebum can be seen on the skin surface after the application of the tonic with *Juniperus communis* fruit oil, whereas the level of sebum on the skin after the application of the tonic with *Pelargonium graveolens* oil remains at a similar level. After 120 min, the amount of sebum on the skin surface after the application of *Pelargonium graveolens* oil is constant, compared to the measurement after 60 min. The amount of sebum on the skin after the application of the tonic with *Juniperus communis* fruit oil increased significantly (up to 101 ± 6.15 a.u.). In summary, both *Pelargonium graveolens* oil and *Juniperus communis* fruit oil have

sebostatic activity at 10 and 60 min after application. However, *Pelargonium graveolens* oil inhibits sebaceous gland activity for a long time, even up to 2 h after application.

*3.5. The Results of Sebum Measurements on the Forehead after Applying the Tonic with 1% and 3% of Niacinamide*

The comparison of average values of sebum measurements after the application of tonics with the addition of 1% or 3% of niacinamide is shown in Figure 3.

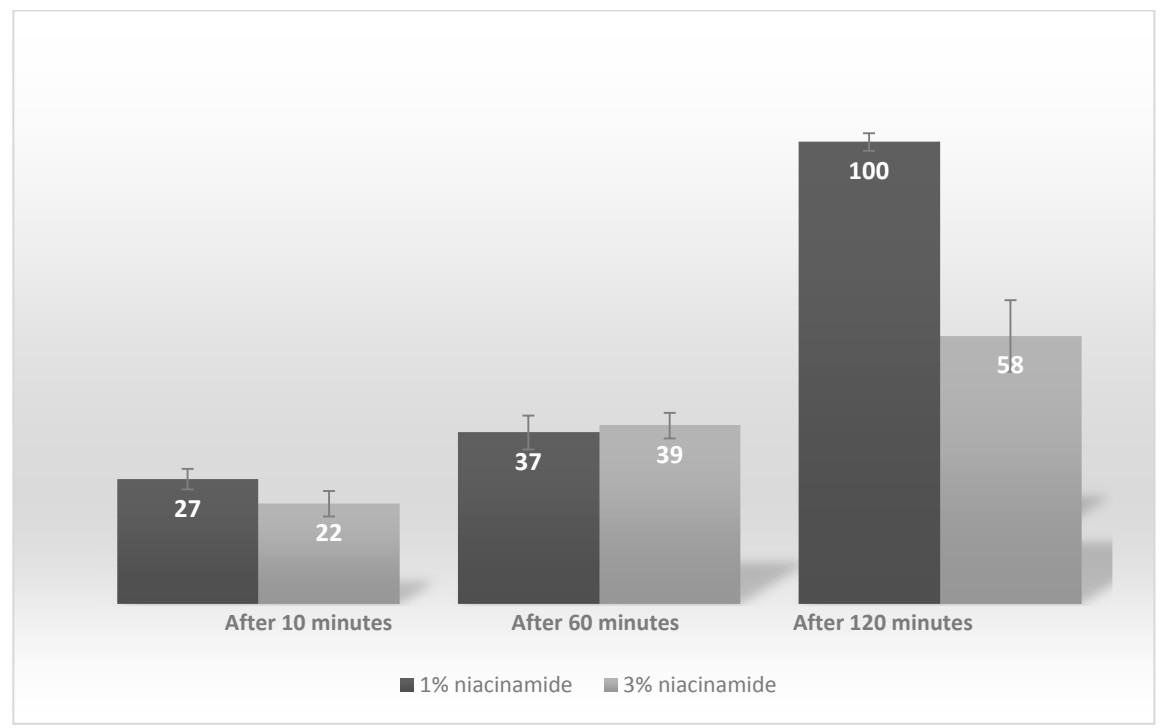

**Figure 3.** Comparison of sebum measurements on the forehead after the application of the tonic with 1% of niacinamide, and the tonic with 3% of niacinamide.

The results indicate that vitamin B3 very effectively reduces the amount of secreted sebum. However, in the case of the tonic containing 1% of niacinamide, this effect can only be observed for 60 min. After 120 min, the level of sebum on the skin increases significantly compared to the level of sebum on the skin 60 min after application.

The addition of 3% vitamin B3 in the cosmetic product effectively reduces the amount of sebum on the skin and maintains the activity of the sebaceous glands at the same level for a long time in comparison to the 1% of added vitamin B3 in the cosmetic product. The amount of secreted sebum on the skin surface after the application of the tonic with 3% of vitamin B3 is maintained at a low level after 10, 60, and 120 min after application and are $22 \pm 2.76$ a.u., $39 \pm 2.78$ a.u., and $58 \pm 9.07$ a.u., respectively.

*3.6. Comparison of Average Values of Sebum Measurements on the Forehead after the Application of the Five Different Tonics*

The comparison of results of sebum measurements on the forehead after the application of all five tonics is shown in Figure 4.

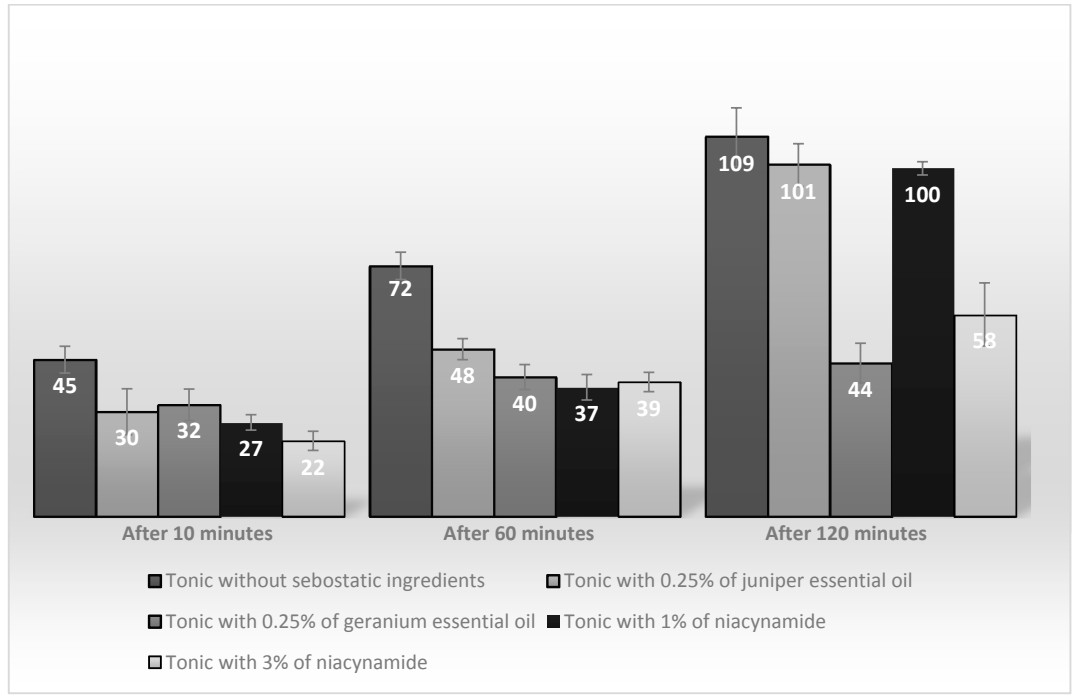

**Figure 4.** Comparison of average values of sebum measurements on the forehead after the application of five different tonics containing *Juniperus communis* fruit oil, *Pelargonium graveolens* oil, niacinamide, or without sebostatic ingredients.

The tonic which most effectively lowers the level of sebum and maintains it at a similar level for the longest period of time is the tonic containing 0.25% of *Pelargonium graveolens* oil in the formulation. The results of sebum measurements on the skin after the application of the tonic containing 0.25% *Pelargonium graveolens* oil change insignificantly over time.

## 4. Conclusions

The evaluation of sebostatic activity of cosmetics for oily skin containing *Juniperus communis* fruit oil, *Pelargonium graveolens* oil, or niacinamide was the purpose of this paper. The tonic with the addition of 3% vitamin B3 and the tonic with the addition of 0.25% *Pelargonium graveolens* oil effectively inhibit the activity of sebaceous glands over a long period of observation.

**Acknowledgments:** Financial support from the National Science Centre (NCN, Poland) Grant No. UMO-2016/21/D/ST8/01705 is gratefully acknowledged.

**Author Contributions:** J.K. conceived and designed the experiments, and supervised the progress the manuscript; A.K. and N.S. performed the experiments and performed the analysis; A.S. critically revised the manuscript.

**Conflicts of Interest:** The authors declare no conflict of interest.

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
