# Peer review of "Evaluation of Sebostatic Activity of Juniperus communis Fruit Oil and Pelargonium graveolens Oil Compared to Niacinamide"

_cosmetics, doi:10.3390/cosmetics4030036_

Round 1

Reviewer 1 Report

The used English language is absolutely not proper for a publication the journal Cosmetics!

The names of plants must be given as follows: Juniperus communis, Pelargonium graveolens, etc.

Different size of words used in the text!

Author Response

Thank you for yours very careful review of our paper and for the comments, corrections and  suggestions that ensued. A revision of the paper has been carried out to take all of them into account. And in the process, we suppose the paper has been significantly improved.  We submit this response letter with responds to comments.

Best regards,

Justyna Kozlowska

The used English language is absolutely not proper for a publication the journal Cosmetics!

*Native language of authors is not English, therefore the authors have sought an advice of a native English speaker. This paper was rechecked carefully.

The names of plants must be given as follows: Juniperus communis, Pelargonium graveolens, etc.

*In this paper, the names of plants were corrected.

Different size of words used in the text!

*It was corrected.

Reviewer 2 Report

This paper is original and can be interesting in cosmetic field. It could be more scietifically supported if more then six patients were tested also because extensive sebum production is a very common condition. Conclusions may point better what's the best sebostatic tonic in short and long obstervation.

Author Response

Thank you for yours very careful review of our paper and for the comments and suggestions that ensued. A revision of the paper has been carried out to take all of them into account. And in the process, we suppose the paper has been significantly improved.  We submit this response letter with responds to comments.

Best regards,

Justyna Kozlowska

This paper is original and can be interesting in cosmetic field. It could be more scientifically supported if more then six patients were tested also because extensive sebum production is a very common condition. Conclusions may point better what's the best sebostatic tonic in short and long observation.

*Thanks a lot for a right comment. Unfortunately, we had a problem with selecting a larger group of participants. We have surveyed about 30 adults to choose participants with similar level of sebum. Unfortunately, only 6 of them were qualified for this study, because the others participants had dry skin or had taken antibiotics, retinoids or hormonal therapy 6 months before testing.

*The part “Conclusions” was rewritten: “The evaluation of sebostatic activity of cosmetics for oily skin containing Juniperus communis fruit oil, Pelargonium graveolens oil or niacinamide was the purpose of this paper. Tonic with 3% addition of a vitamin B3 and tonic with 0.25% addition of Pelargonium graveolens oil effectively inhibit the activity of sebaceous glands in long observation.”

Round 2

Reviewer 1 Report

All necessary corrections and changes from the original manuscript have been accepted in the revised one!